# Benzodiazepines Drive Alteration of Chromatin at the Integrated HIV-1 LTR

**DOI:** 10.3390/v12020191

**Published:** 2020-02-09

**Authors:** Weam Elbezanti, Angel Lin, Alexis Schirling, Alexandria Jackson, Matthew Marshall, Rachel Van Duyne, Frank Maldarelli, Luca Sardo, Zachary Klase

**Affiliations:** 1Department of Biological Sciences, University of the Sciences, Philadelphia, PA 19104, USA; 2HIV Dynamics and Replication Program, Center for Cancer Research, National Cancer Institute, Frederick, MD 21702, USA; 3Currently at Department of Infectious Diseases and Vaccines, MRL, Merck & Co., Inc., West Point, PA 19486, USA; 4Substance Use Disorder Institute, University of the Sciences, Philadelphia, PA 19104, USA

**Keywords:** HIV-1 transcription, viral transcription, benzodiazepines, chromatin, alprazolam, STAT5, RUNX1

## Abstract

Antiretroviral therapy (ART) lowers human immunodeficiency virus type 1 (HIV-1) viral load to undetectable levels, but does not eliminate the latent reservoir. One of the factors controlling the latent reservoir is transcriptional silencing of the integrated HIV-1 long terminal repeat (LTR). The molecular mechanisms that control HIV-1 transcription are not completely understood. We have previously shown that RUNX1, a host transcription factor, may play a role in the establishment and maintenance of HIV-1 latency. Prior work has demonstrated that inhibition of RUNX1 by the benzodiazepine (BDZ) Ro5-3335 synergizes with suberanilohydroxamic acid (SAHA) to activate HIV-1 transcription. In this current work, we examine the effect of RUNX1 inhibition on the chromatin state of the integrated HIV-1 LTR. Using chromatin immunoprecipitation (ChIP), we found that Ro5-3335 significantly increased the occupancy of STAT5 at the HIV-1 LTR. We also screened other BDZs for their ability to regulate HIV-1 transcription and demonstrate their ability to increase transcription and alter chromatin at the LTR without negatively affecting Tat activity. These findings shed further light on the mechanism by which RUNX proteins control HIV-1 transcription and suggest that BDZ compounds might be useful in activating HIV-1 transcription through STAT5 recruitment to the HIV-1 LTR.

## 1. Introduction

The major challenge for human immunodeficiency virus type 1 (HIV-1) eradication is the viral reservoir harbored by resting memory CD4^+^ T cells that is established early during infection [1,2,3]. In these cells, integrated proviral genomes undergo transcriptional silencing, can evade the immune system, are not able to be acted upon by traditional therapies, and allow the virus to persist for many years [1,2,4,5]. Upon cessation of therapy, the virus can be reactivated within the patient’s body [6]. Recent studies propose a way to eradicate HIV-1 through activation of the silenced virus in resting memory CD4^+^ cells through a “shock and kill” approach [7,8]. In this approach, the HIV-1 provirus would be activated using latency reversing agents (LRAs) and infected cells would then be depleted by viral cytopathic effects, the host immune response or a second therapeutic intervention. At the same time, new infection of other cells would be controlled using antiretroviral therapy (ART) [9]. Proposed LRAs work through interfering with cellular mechanisms known to be involved in maintaining HIV-1 in a transcriptionally silent state [10]. The current commonly targeted mechanisms include (i) histone deacetylase inhibitors (HDACi) such as panobinostat, suberanilohydroxamic acid (SAHA; Vorinostat) and Trichostatin-A (TSA); (ii) bromodomain-containing protein inhibitors such as JQ1, and (iii) protein kinase C (PKC) agonists [11], which work through activation of NF-κB such as prostratin, bryostatin and ingenol. SAHA and other treatments have been unsuccessful in decreasing the size of the latent pool in vivo [9,12,13,14,15]. Therefore, there is a need to further characterize the mechanisms that control HIV-1 transcription. 

The HIV-1 long terminal repeat (LTR) promoter is regulated by numerous transcription factors and chromatin-associated proteins. We have previously shown that the U3 region of HIV-1 contains a potential binding site for RUNX1 protein and that overexpression of RUNX1 and/or core binding factor-β (CBF-β) inhibits HIV-1 transcription. Conversely, suppression of endogenous RUNX1 or CBF-β with siRNA significantly increases HIV-1 transcription [16]. The benzodiazepine (BDZ) inhibitor of RUNX1/ CBF-β function, Ro5-3335 [17], synergizes with the HDAC inhibitor SAHA in activating HIV-1 transcription [16]. RUNX1 is one of three RUNX proteins found in humans, all of which possess a highly conserved 128 amino acid Runt DNA-binding domain at the N-terminus. The Runt domain is important for binding to the DNA consensus sequence (PyGPyGGTPy) and to the core binding factor subunit beta (CBF-β) [18]. CBF-β heterodimerizes with RUNX1 [19], enhances RUNX affinity for the DNA by 10 fold [20] and stabilizes RUNX proteins against proteolytic degradation [21]. RUNX proteins contain a nuclear localization signal that allows the transport of the CBF-β-RUNX complex to the nucleus and regulate gene expression [22,23]. The RUNX proteins are highly similar and share a common DNA binding site. However, they exert different physiologic functions [24].

According to its posttranslational modifications and the context in which it is recruited, RUNX1 functions to activate or repress gene transcription. It can initiate transcription through recruitment of the CBP/P300 histone acetyltransferase [25]. On the other hand, it may repress transcription through recruitment of repressive factors such as mSin3A, SUV39H1 and histone deacetylases [26,27]. Furthermore, RUNX1 physically interacts with the STAT5 transcription factor and this interaction regulates either RUNX1 or STAT5 function [28]. The phosphorylation of STAT5 by JAK kinases has been shown to induce recruitment of STAT5 to the integrated HIV-1 LTR and activation of HIV-1 transcription [29,30]. 

In this work, we examine the molecular mechanism by which RUNX1 alters HIV-1 transcription. We show that RUNX inhibitors aid in the recruitment of STAT5 to the HIV-1 promoter. We also demonstrate that the previously observed inhibition of Tat transactivation by Ro5-3335 is not tied to the ability to positively regulate transcription. We show that other BDZ compounds can positively regulate transcription without suppressing Tat. We further examine the mechanism of transcriptional increase by the BDZ alprazolam and show that alprazolam has an effect on the recruitment of positive transcription factors to the integrated LTR. 

## 2. Materials and Methods

### 2.1. Cell Culture and Transfections 

J-Lat Full Length Cells (10.6) (NIH-AIDS Reagent Program, catalog number 9849, NIH, Bethesda, MD, USA) were maintained in Roswell Park Memorial Institute medium 1640 (RPMI; HyClone, GE Healthcare Life Sciences, Chicago, IL, USA) supplemented with 10% fetal bovine serum (FBS; HyClone, GE Healthcare Life Sciences, Chicago, IL, USA), 100 units mL^−1^ penicillin,100 µg mL^−1^ streptomycin and 0.3 mg mL^−1^ L-glutamine (PSG; Gibco, Fisher Scientific, Gaithersburg, MD, USA). Cells were grown at 37 °C with 5% CO_2_. TZMbl cells (NIH-AIDS Reagent Program, catalog number 8129, NIH, Bethesda, MD, USA) and 293T cells (ATCC, catalog number CRL-11268, Manassas, VA, USA) were maintained in Dulbecco’s modified Eagle’s medium (DMEM; HyClone, GE Healthcare Life Sciences, Chicago, IL, USA) supplemented with 10% FBS, 100 units mL^−1^ penicillin, 100 µg mL^−1^ streptomycin and 0.3 mg mL^−1^ L-glutamine. Cells were grown at 37 °C with 5% CO_2_. To transfect TZMbl with a flag-Tat expression construct, Lipofectamine LTX (Invitrogen, Carlsbad, CA, USA) was used according to the manufacturer’s instructions.

Peripheral blood mononuclear cells (PBMCs) were purified from whole blood samples using Ficoll (GE Healthcare Life Sciences, Chicago, IL, USA) gradient centrifugation and cryopreserved in 90% FBS containing 10% dimethyl sulfoxide (DMSO; Fisher Scientific, Gaithersburg, MD, USA). PBMCs were thawed and cultured in RPMI complete media (RPMI 1640 media supplemented with 10% FBS and 100 units mL^−1^ penicillin,100 µg mL^−1^ streptomycin and 0.3 mg mL^−1^ L-glutamine.

FDA approved BDZs are from (Sigma-Aldrich, St. Louis, MO, USA): Alprazolam (A0357000), Bromazepam (pB4144), Clobazam (C8414), Clonazepam (C1277), Clorazepate (1140509), Diazepam (D0940000), Estazolam (E3638) and Flunitrazepam (F9261). The LRA compound SAHA was also obtained from Sigma-Aldrich (SML0061).

For determination of HIV-1 LTR activation in TZMbl cells, cultures were treated with BDZs or SAHA for 48 h. Following treatment, cell lysates were prepared using GloLysis buffer (Promega, Madison, WI, USA) and luciferase activity was determined using BrightGlo Luciferase Reagent (Promega, Madison, WI, USA) and read on a spectrophotometer following manufacturers’ instructions. 

For examination of non-HIV promoters, 293T cells were plated at 2 × 10^5^ cells/well of a 12-well plate. One day after plating, the cells were transfected with pUC19, pHIV-LTR-GL3 WT, pHTLV-LTR-Luc [31] or pNF-κB-Luc (Stratagene, San Diego, CA, USA) using Lipofectamine 3000 (Invitrogen, Carlsbad, CA, USA) according to manufacturer’s instructions with the indicated plasmids. At 24 h post-transfection, cells were removed from the plate with cell dissociation buffer (5mM EDTA + PBS) and replated at ~1.6 × 10^4^ cells/well of a white, clear-bottomed 96-well plate and treated with Alprazolam (10μM). Approximately 30 h post-treatment, luciferase activity was determined using BrightGlo Luciferase Reagent and read as above.

### 2.2. Flow Cytometry and Primary Cell Analysis

For the activation work, J-Lat 10.6 cells were treated with different concentrations of indicated FDA BDZs (0.01 μM, 0.1 μM, 1 μM, and 10 μM). Cells were analyzed for cell death using Zombie Yellow Fixable Live/Dead stain (77168, BioLegend, San Diego, CA, USA) and green fluorescent protein (GFP) expression was measured using flow cytometry. To examine safety of the treatment and effect on T cell activation and HIV activation in primary cells, we treated JLat 10.6 with different BDZs in the presence or absence of SAHA. At 48 h after incubation, cells were harvested and stained with Live/Dead^®^ Fixable Red Dead Cell Stain Kit (L34957, Life Technologies, Carlsbad, CA, USA) staining according to the manufacturer’s protocol. To test T cell activation in primary cells, PBMC from healthy donors were cultured overnight in the presence or absence of Alprazolam, SAHA, Ro5-3335 or in combination. Cells then stained using Zombie Yellow Live/Dead staining and then stained for CD3 (BDB557917), CD8 (BDB563823), CD4 (BDB552838), CD69 (BDB557745) (BD Biosciences, San Jose, CA, USA), and then fixed for analysis by flow cytometry (Fisher Scientific, Gaithersburg, MD, USA). Populations of T cells from each treatment were used to prepare RNA using Trizol reagent following the manufacturer’s protocol (Thermo Fisher Scientific, Waltham, MA, USA). Following reverse transcription, the samples were diluted 1:50, and 2.5 microliters were used for quantitative PCR in a BioRad CFX96 or CFX384 qPCR machine (Hercules, CA, USA). RT-qPCR for HIV-1 Gag was normalized to GAPDH. 

The following primer pairs were used for detection: Gag: 5′ GGTGCGAGAGCGTCAGTATTAAG 3′, 5′ AGCTCCCTGCTTGCCCATA 3′GAPDH: 5′ GCTCACTGGCATGGCCTTCCGTGT 3′, 5′ TGGAGGAGTGGGTGTCGCTGTTGA 3′

Expected size of Gag PCR fragment is 119nt; GAPDH is 200nt.

### 2.3. Nuclei Isolation and Chip Assay

TZMbl cells were cultivated for 48 h with media supplemented with Ro5-3335 and SAHA. Then, cells were harvested by trypsinization and nuclei isolated by chemical lysis and stored following the procedure for cell suspension lines recommended by the manufacturer—Nuclei EZ Prep (Nuc-101, Sigma-Aldrich, St. Louis, MO, USA). Nuclei were stained for 1h on ice in 5% FBS in 1× -phosphate-buffered saline (5% FBS/PBS) solution containing primary antibodies. Nuclei were washed twice in 5% FBS/PBS and stained for 1h on ice with appropriate secondary antibodies [Alexa Fluor 555 anti-rabbit IgG (A21429), Alexa Fluor 488 anti-mouse IgG (A11001); Thermo Fisher Scientific, Waltham, MA, USA) conjugated with fluorescent fluorophores. Following two washes in 5% FBS/PBS, nuclei were resuspended in the storage buffer provided by the kit (Nuclei EZ Prep). Nuclei were then imaged using a Zeiss Axiovert (Jena, Germany) microscope with a Zeiss alpha Plan-Fluar 100×, 1.45 NA oil objective and a spinning disk confocal scan head (Yokogawa CSU-X1, Tokyo, Japan). Computational image analysis of the fluorescent signals was performed using the Localize software package from Daniel Larson (NIH, Bethesda, MD, USA).

For chromatin immunoprecipitation (ChIP) analysis, TZMbl were treated with indicated compounds and fixed according to manufacturer’s protocol: Pierce Agarose ChIP kit (26156, Thermo Fisher Scientific, Waltham, MA, USA). All antibodies used were diluted 1:100, and included: Histone H3 (acetyl K9) antibody [AH3-120]-ChIP Grade (ab12179, Abcam, Cambridge, UK), Pierce p300/CBP (CREB-Binding Protein) antibody (NM11) (MA5-13634, Thermo Fisher Scientific, Waltham, MA, USA), Human/Mouse STAT5a/b Pan Specific Antibody (AF2168, R&D Systems, Minneapolis, MN, USA), RUNX1/AML1 (ab23980, Abcam, Cambridge, UK), CBF-β (ab167382, Abcam, Cambridge, UK). The following primer pair against the HIV U3 LTR used for qPCR was as follows:

5’ CTAGCATTTCGTCACATGGCCCGAGA 3’, 5’ GTGGGTTCCCTAGTTAGCCAGAG 3’

### 2.4. Statistical Analysis 

Data represented in graphical form is always the average of at least three replicates with error bars on the graph representing standard deviation. Statistical significance was determined using an unpaired Student’s t-test comparing a given condition to control * P ≤ 0.05, ** P ≤ 0.01, *** P ≤ 0.001.

## 3. Results

### 3.1. RUNX1 Inhibition Drives Chromatin Changes at the HIV-1 LTR

Our previous work demonstrated that inhibition of RUNX1 by Ro5-3335 improved on the ability of SAHA to turn on HIV-1 LTR driven transcription in patient samples and a variety of cell lines [16]. To further characterize the effect of Ro5-3335 treatment in the presence of SAHA, we performed chromatin immunoprecipitation (ChIP) to examine the presence of histone modifications and transcription factors at the integrated HIV-1 LTR. For these assays, we chose the TZMbl reporter cell line that contains an integrated HIV-1 LTR driving expression of firefly luciferase and a second integrated HIV-1 LTR driving the β-galactosidase reporter gene. Use of this simplified promoter system allowed us to examine changes at the integrated promoter in the absence of the viral transactivator Tat and other viral proteins. TZMbl cells were treated for 48 h with or without SAHA and Ro5-3335 before being harvested for ChIP using antibodies that recognize acetylated histone H3 (H3K9ac, a known marker for transcriptional activation [32,33]), RUNX1, CBF-β and STAT5 (Figure 1). qPCR was used to determine occupancy of these proteins at the integrated LTR. SAHA at 0.25 or 1µM caused relatively little increase in the acetylation of histone H3 compared to the DMSO control. This is in line with the concentration of SAHA needed to effectively activate this model system (Figure 2A). Addition of 50µM Ro5-3335 to cells treated with 1µM SAHA resulted in a notable increase in acetylation compared to 1µM SAHA alone (Figure 1A). Interestingly, SAHA treatment alone actually increased the presence of RUNX1 and its binding partner CBF-β at the integrated LTR compared to the DMSO control (Figure 1B,C). Our prior work on the involvement of RUNX1 in transcriptional control of latent HIV-1 showed that SAHA increased the gene expression of RUNX1 [16]. This potential increase in RUNX1 expression might explain the observed synergy between SAHA and the RUNX1 inhibitor Ro5-3335. Using a new imaging technique that we have recently developed called NUCLEO-M, which allows fluorescent imaging of intact, unfixed nuclei [34], we verified these results at the protein level (Figure 2B,C). As expected, treatment of cells with SAHA lead to an increase in total acetylated H3K9 in the nuclei. Treatment with Ro5-3335 and SAHA further increased the acetylation in keeping with previously observed increases in transcription. Treatment with SAHA alone also increased the total presence of RUNX1 in the nuclei as expected. Inhibition of RUNX1 function by Ro5-3335 lead to a decrease in RUNX1 both specifically interacting with the integrated LTR (Figure 1B) and expression levels throughout the nucleus (Figure 2B,C). This is likely due to the presence of RUNX binding sites in the RUNX1 promoter that respond positively to RUNX1 binding [35]. Inhibition of RUNX1 by Ro5-3335 may block this activation and lead to the observed decrease in RUNX1 expression. Finally, RUNX1 has been demonstrated to suppress the function of the transcription factor STAT5 [28] that has been shown to have a positive effect on HIV-1 transcription [36]. Therefore, we performed ChIP for STAT5 to determine if Ro5-3335 treatment might increase the recruitment of STAT5 (Figure 1D). STAT5 occupancy on the HIV-1 LTR was increased in the presence of Ro5-3335 and SAHA compared to SAHA alone. These results suggest that RUNX1 inhibition with Ro5-3335 serves to increase SAHA driven activation of the LTR by limiting the effect of RUNX1 at the LTR and by driving recruitment of STAT5—a function that SAHA alone does not perform. 

### 3.2. Screening of Benzodiazepines (BDZs) for Improved Potency 

The benzodiazepine Ro5-3335 was described ~20 years ago to be an inhibitor of HIV-1 Tat activity [37]. A small clinical trial of its analog, Ro24-7429, determined it not to be an effective intervention for acute infection [38]. Work by the Liu group identified the RUNX suppressive activity of Ro5-3335 and showed that the drug interacted with the HIV-1 Tat protein [17]. We were curious if Ro5-3335 would suppress Tat transactivation in our system. For these studies, we transfected TZMbl cells in 96-well format with a plasmid encoding HIV-1 Tat, treated 24 h later with or without Ro5-3335 and then determined luciferase activity 24 h after treatment (Figure 3). As expected, Ro5-3335 significantly suppressed LTR-driven luciferase expression in keeping with its description as a Tat inhibitor.

We next sought to determine if other BDZ compounds might be able to more potently activate HIV-1 transcription while avoiding Tat suppression. BDZs have been used for many decades for controlling anxiety, depression, convulsion and sleeping disorders [39,40,41]. This means that a large number of well characterized compounds exist. We chose eight of the FDA approved BDZs to screen for the ability to activate the HIV-1 LTR (Table 1). For screening we used the J-Lat 10.6 T-cell line containing an integrated HIV-1 provirus (single cycle) that encodes GFP. Transcriptional activation was measured by flow cytometry as the percentage of GFP+ live cells 48 h following treatment with 10 µM BDZs; TSA, a non-specific inhibitor of class I and II HDACs, was used as a positive control for activation (Figure 4A). Treatment with DMSO showed only a background level of GFP+ cells (2.8%). Similarly, Ro5-3335 alone did not significantly activate transcription above background (3.0%). Of all of the BDZs tested, Alprazolam and Diazepam treatment resulted in significant activation of the LTR compared to the DMSO control (20.3% and 12.4% respectively); the remainder of the BDZs induced small, but occasionally statistically significant changes in the percentage of GFP+ cells. Treatment of J-Lat 10.6 cells with 0.5 µM SAHA induced a minor increase in GFP+ cells compared to the DMSO control (6.2%). When cells were treated with BDZs in combination with SAHA, we observed a significant increase in GFP+ cells compared to SAHA alone. We also observed an increase in LTR activation in cells treated with BDZs and SAHA compared to cells treated with BDZs alone with the exception of Alprazolam and Diazepam which already showed maximum activation of the assay in the absence SAHA. Cell viability of cells from Figure 4A was determined 48 h after treatment; cells were harvested and stained with Live/Dead staining according to the manufacturer’s protocol and measured by flow cytometry as the percentage of live cells (Figure 4B). TSA induced significant toxicity (68% viable) whereas 0.5 μM SAHA resulted in only a slight reduction in viability compared to DMSO control. No significant loss of viability was measured in the presence of BDZs and treatment with SAHA and BDZs showed no decrease in viability beyond SAHA alone. 

To further investigate the effect of BDZs on the activation of the HIV-1 LTR in the presence or absence of SAHA, we treated J-Lat 10.6 cells with increasing concentrations of Alprazolam, Clonazepam and Clorazepate in the presence or absence of 0.5 μM SAHA (Figure 4C). We again measured an increase in GFP+ cells in the presence of Alprazolam at 10 μM (25.5% compared to 2.8% for the DMSO control) in the absence of SAHA (Figure 4C, left panel). All doses of Alprazolam tested showed a significant increase in activation of the LTR compared to DMSO alone. Treatment of cells with 0.1 μM Alprazolam in the presence of 0.5 μM SAHA showed an appreciable increase of activation over SAHA alone (13.7% compared to 5.9%, respectively). Clonazepam was able to increase activation in the presence of SAHA only at concentrations of 1 μM and 10 μM (Figure 4C, center panel). Clorazepate alone demonstrated significant activity over the DMSO control at 100 nM (9.7% compared to 5.6%, respectively) and at 1 μM (8.9% compared to 5.6%, respectively) (Figure 4C, right panel). Clorazepate was also capable of increasing activation in the presence of SAHA compared to SAHA alone at all concentrations tested (for example, at 100 nM Clorazepate 20.3% compared to 12.7%). Analysis of the effect of Alprazolam on plasmid based HIV-1 LTR, HTLV LTR and NF-κB reporters suggested that the observed increase in transcription is specific to the HIV-1 LTR (Appendix A). Given the robust ability to activate the HIV-1 LTR in the presence and absence of SAHA, we elected to focus on Alprazolam.

To investigate whether Alprazolam is specifically activating the HIV-1 promoter or having a general effect on T cell activation, primary peripheral blood mononuclear cells (PBMCs) from two healthy donors were treated with 10 μM alprazolam or 50 μM Ro5-3335 alone or in combination with 0.5 μM SAHA. At 24 h after treatment, cells were stained using Zombie Yellow Live/Dead stain, then stained for CD3, CD4, CD8, HLA-DR, CCR5, intracellular Ki67, and the early T-cell activation marker CD69, and then fixed for analysis by flow cytometry (Figure 5A,B; Appendix A). Anti CD3/CD28 (data not shown) and IL2/phytohaemagglutinin (PHA) treatment were included as a positive control for T-cell activation. These treatments did not cause an appreciable loss of cell viability (Figure 5A) and Alprazolam treatment in combination with SAHA did not appreciably increase T cell activation (Figure 5B). Examination of intracellular Ki67, CCR5 and HLA-DR revealed that neither SAHA, Alprazolam, nor a combination of the two induced an upregulation of these activation markers (Appendix A). 

To evaluate the ability of Alprazolam to induce transcription of integrated LTR promoters in cells from HIV-1 infected individuals, we obtained PBMCs from two donors who had been suppressed on ART for greater than six months. 10 × 10^6^ PBMCs were divided between three conditions: DMSO control, 50 μM Ro5-3335 and 10 μM alprazolam. At 24 h post-treatment, RNA was extracted from the cells and used for RT-qPCR to detect HIV-1 Gag mRNA (Figure 5C). As previously shown, Gag RNA was detectable at low levels in unstimulated PBMCs from these patients. As expected, the use of Ro5-3335 alone had a negligible effect on HIV-1 transcripts (1.4 and 1.1-fold increase over DMSO control in Patients 1 and 2, respectively). Alprazolam induced a larger fold increase in Gag RNA in these samples (4.6 and 3.2-fold in Patients 1 and 2, respectively). This limited analysis suggests that alprazolam may increase transcription of HIV-1 in the cells of infected individuals.

To confirm the observed effect of Alprazolam activation of HIV-1 LTR transcription in a minimal system and to determine the effect on Tat-activated transcription we again employed the TZMbl cell line. TZMbl cells were treated with Ro5-3335, Alprazolam or Clonazepam in the presence or absence of SAHA (Figure 6A,B) and LTR-driven luciferase activity was determined at 48 h post treatment. Alprazolam (10 μM) alone was capable of significantly inducing transcription from the LTR and the addition of SAHA further increased activation (Figure 6B). To reinforce the specificity of Alprazolam treatment on Tat activated transcription we transfected TZMbl cells in 96-well format with a plasmid encoding HIV-1 Tat, treated 24 h later with Ro5-335, Alprazolam, or Clonazepam and then determined activation by luciferase assay 24 h post treatment (Figure 6C). As expected, Ro5-3335 significantly suppressed Tat transactivation compared to the DMSO control. Alprazolam and Clonazepam did not exhibit an inhibition of Tat-mediated activation of the LTR. These data suggest that the mechanism of action of activation of the LTR by Alprazolam is distinct from that of Tat. Alprazolam does not act as an inhibitor of Tat, differing significantly from Ro5-3335.

### 3.3. Epigenetic Changes Driven by Alprazolam at the HIV-1 LTR 

The ability of Alprazolam to induce transcription from the HIV-1 LTR suggests that it may have a different mechanism of action than Ro5-3335 and the other BDZs tested. In order to query the changes driven by these compounds, we again performed ChIP on TZMbl cells to examine the presence of various transcription factors and post-translational modifications (PTMs) on the integrated HIV-1 LTR. TZMbl were treated for 48 h with Ro5-3335, Alprazolam or Clonazepam in the presence or absence of SAHA. We first examined the presence of H3K9ac and, as expected, treatment with SAHA increased the occupancy of H3K9ac at the promoter as compared to DMSO control (Figure 7A). Alprazolam, but not Clonazepam or Ro5-3335, further increased H3K9ac compared to SAHA alone. Treatment with BDZs alone were not sufficient to alter H3K9ac. We also performed ChIP for STAT5 to determine if BDZ treatment might increase the recruitment of STAT5 as expected of a RUNX inhibitor [28] (Figure 7B). SAHA, Ro5-3335 and Clonazepam all caused a small, but statistically significant drop in STAT5 occupancy at the LTR as compared to DMSO control. Alprazolam showed a >8 fold increase in STAT5 recruitment to the LTR. In combination with SAHA, all three BDZs were significantly more capable of driving recruitment of STAT5 to the promoter as compared to BDZs alone. STAT5 is capable of recruiting the histone acetyltransferase CBP/P300 that acetylates histone 3 at lysine 27 (H3K27ac) [50,51]. Additionally, this histone acetyltransferase has been shown to have activity in acetylating HIV-1 Tat protein and modulating its activity [51,52,53]. ChIP for CBP/P300 showed no increase in recruitment upon treatment with SAHA alone compared to DMSO (Figure 7C). The addition of both Ro5-3335 or Alprazolam to SAHA treatment induced recruitment of CBP/P300 to the HIV-1 LTR (by three and seven fold respectively). The efficient recruitment of STAT5 and CBP/P300 may explain the ability of Alprazolam to increase HIV-1 transcription. 

## 4. Discussion

In our previous study, we showed that the BDZ inhibitor of RUNX1, Ro5-3335, synergizes with SAHA to induce HIV-1 transcription [16]. Here, we examined the molecular mechanisms by which RUNX1 inhibition may alter HIV-1 transcription. Using ChIP assays, we were able to show that Ro5-3335 can increase SAHA driven acetylation of histones and may drive recruitment of STAT5 to the LTR (Figure 1). We also noted that both overall levels of RUNX1 in the nucleus and specifically recruited to the LTR were increased by exposure to SAHA. This effect was reversed with the addition of Ro5-3335. This is likely due to the ability of RUNX1 to positively regulate its own promoter; therefore, inhibition of RUNX1 protein function reduces RUNX1 mRNA expression [35]. This observation may explain why RUNX1 inhibition by BDZs potentiates acetylation by SAHA.

We also confirmed previous reports that Ro5-3335 can counteract Tat transactivation of the HIV-1 LTR (Figure 3). As such, we sought to determine if it was possible to decouple pharmacologic inhibition of RUNX1 from Tat inhibition by testing a battery of eight different BDZs (Table 1). Interestingly, all of the BDZs tested caused an increase in transcription as measured by GFP+ cells when used in combination with SAHA compared to BDZ treatment alone (Figure 4A). At higher concentrations, Alprazolam induced maximal activation of the LTR without SAHA treatment. When a dose range was tested, all doses of Alprazolam showed an increase in activation and an improvement over SAHA. We found that Clonazepam and Clorazepate also induced a significant increase in the HIV-LTR driven gene expression in combination with SAHA. Neither Alprazolam, nor Clonazepam were capable of suppressing Tat activity.

In order to elucidate the mechanistic differences between Ro5-3335 or Clonazepam and Alprazolam we performed ChIP assays to compare the effect of the two BDZs on the chromatin structure of the HIV-1 LTR. We limited our search to RUNX1 related factors and the H3K9 acetylation mark. ChIP analyses (Figure 7A), showed that BDZs do not affect acetylation by themselves. Alprazolam, but not Ro5-3335 or Clonazepam, was able to recruit STAT5 by itself. Recruitment of CBP/P300 by STAT5 and subsequent acetylation of H3K27 may explain the functional difference between these BDZs (Figure 8). Alprazolam also potentiated SAHA’s ability to drive H3K9ac occupancy at the HIV-1 LTR. This suggests that these BDZs may have an indirect effect on acetylation of H3K9. One of these may be the inhibition of RUNX1, which recruits HDACs to silence some genes. RUNX1 is also involved in recruiting polycomb repressive complex 1 (PRC1) in megakaryocytes and lymphocytes [54], which has been shown to be overexpressed in HIV-1 latent cell models [55]. PRC1 can recruit polycomb repressive complex 2 (PRC2) [56], a methyltransferase that targets H3K27 for mono-, di-, and tri- methylation. H3K27 is highly methylated in silenced HIV-1 chromatin [57]. H3K27 demethylation has been shown to augment the ability of SAHA to activate the latent reservoir [58]. Further work must be done to identify what effect BDZs may have on H3K27 and how they function in the presence of other transcriptional activators.

RUNX1 has been shown to suppress the transcriptional activity of STAT5 [28], which is known to activate HIV-1 transcription [36]. We performed ChIP to investigate whether BDZs increase the level of STAT5 at the HIV-1 promoter (Figure 7B). Treatment with Ro5-3335, Alprazolam or Clonazepam in combination with SAHA significantly increased STAT5 levels at HIV-1 promoter compared to BDZs without SAHA. Of the BDZs tested, only treatment with Alprazolam showed a significant recruitment of STAT5 when used alone. Tat acetylation by CBP/P300 enhances HIV-1 transcription and its binding to core histones [52]. CBP/P300 can be recruited by STAT5. We tested the effect of BDZs on the level of the CBP/P300 at the HIV-1 LTR (Figure 7C). There was a significant increase of CBP/P300 at the HIV-LTR when Alprazolam and Ro5-3335 were added to SAHA. Whether this influences Tat acetylation is not yet known.

It is also possible that Alprazolam may exert its effects through other interactions. Alprazolam contains a unique chemical structure which includes the benzene ring fused to diazepine and a triazole ring (8-chloro-1-methyl-6-phenyl-4H-[1,2,4]triaz-olo[4,3-a][1,4]benzodiazepine). It has been shown by Bosque et al. that benzotriazoles increased STAT5 activity, nuclear localization, and its occupancy at the HIV-LTR through hindering its SUMOylation step [29]. Alprazolam contains a 3-methyl group attached to Triazolo ring that has been shown to be mediate interaction with and inhibition of BET bromodomains (BET BRDs), especially BRD4 [59]. BRD4 antagonizes Tat interaction with positive transcription elongation factor complex (P-TEFb), which is an important transcription factor involved in HIV transcription [60]. Competition with Tat is the mechanism that is proposed for JQ1, a prototype BDZ molecule that inhibits BRD4 and reactivates HIV-1 [61,62]. Although our studies began with a focus on RUNX1, the data presented do not allow us to determine if observed changes are due to RUNX1 inhibition or direct STAT5 interaction. Our data does not allow us to conclude that BDZs block all RUNX1 function, but suggest that they favor the recruitment of positive factors by RUNX1 and alter chromatin structure of the LTR to induce transcription.

## 5. Conclusions

These findings provide further support for the role of RUNX proteins in HIV-1 transcriptional control. Based on observations in this paper we propose that various benzodiazepines may act as RUNX1 inhibitors and through RUNX1 inhibition drive STAT5 recruitment and changes at the integrated LTR that lead to an increase in transcription.

## Figures and Tables

**Figure 1 viruses-12-00191-f001:**
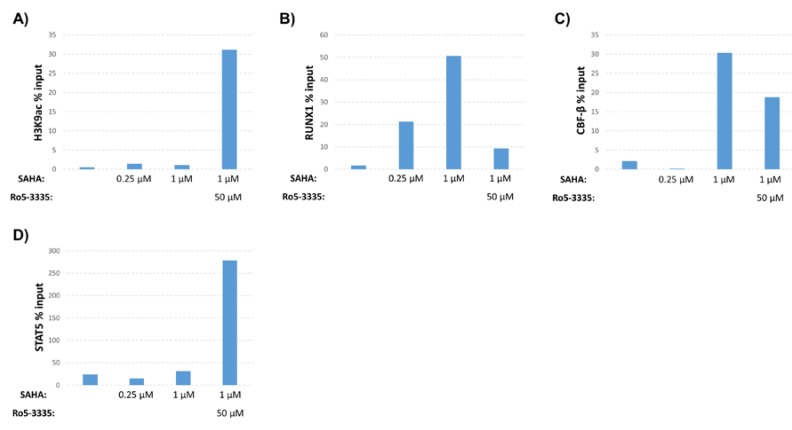
Effect of suberanilohydroxamic acid (SAHA) and Ro5-3335 treatment on the human immunodeficiency virus type 1 (HIV-1) long terminal repeat (LTR). TZMbl cells were cultured for 48 h in medium containing indicated concentrations of SAHA in the presence or absence of 50μM of Ro5-3335. Cells were harvested for chromatin immunoprecipitation (ChIP) using antibodies against (**A**) histone H3 acetylated at lysine 9 (H3K9ac), (**B**) RUNX1, (**C**) core binding factor-β (CBF-β) or (**D**) STAT5.

**Figure 2 viruses-12-00191-f002:**
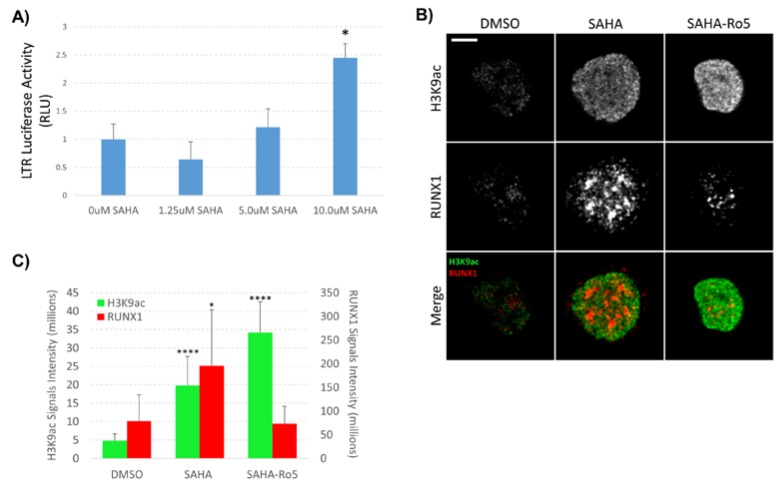
Effect of SAHA and Ro5-3335 treatment on RUNX1 expression in the nucleus. (**A**) LTR-driven luciferase activity was determined post-treatment with the indicated concentrations of SAHA; (**B**) nuclei were isolated from TZMbl cells treated with 1µM SAHA and 50µM Ro5-3335 and stained for H3K9ac (green) and RUNX1 (red). Spinning disk confocal microscopy was used for imaging using 100X oil objective. Images of bright field and nuclear fluorescence were captured using a high-speed camera. (**C**) Localize software was used to measure signal intensity in the collected images. Scale bars: 5 μm. * *P* ≤ 0.05, *** *P* ≤ 0.001.

**Figure 3 viruses-12-00191-f003:**
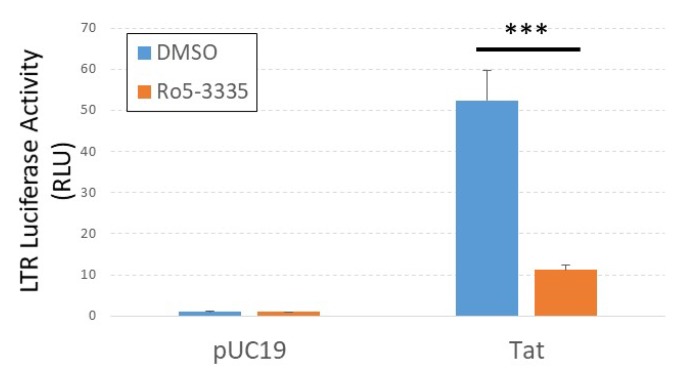
Ro5-3335 inhibits Tat transactivation of the integrated LTR. TZMbl cells were transfected with pUC19 control plasmid or pCMV-Tat plasmid, treated with 50μM of Ro5-3335 at 24 h post-transfection, and measured for luciferase expression 48 h post-transfection. *** *P* ≤ 0.001.

**Figure 4 viruses-12-00191-f004:**
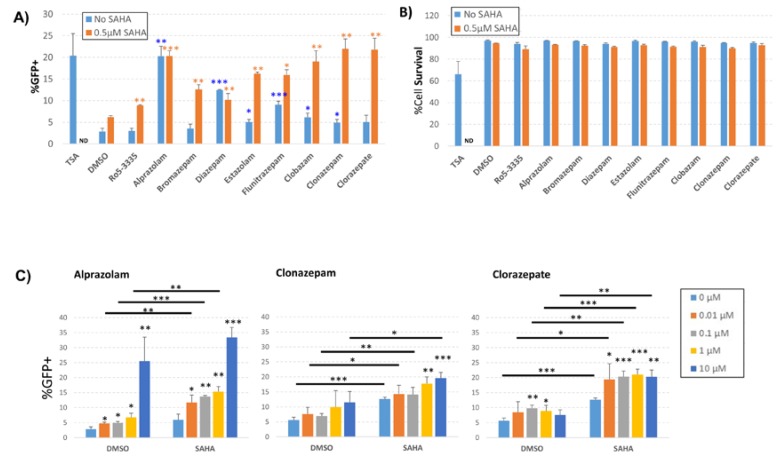
Effect of benzodiazepines (BDZs) on transcription in J-Lat 10.6 cells. J-Lat 10.6 cells were cultured with the indicated BDZs in the presence or absence of 0.5 μM SAHA. 48 h after treatment, cells were measured for the percentage of GFP positive cells and were stained for viability as determined by flow cytometry. (**A**) The percentage of GFP positive live cells after treatment with 10 μM of different BDZs. (**B**) 48 h after treatment, cells were stained for live/dead determination and the percentage of live cells was measured by flow cytometry. (**C**) Dose response graph for cells treated with selected BDZs alone or in combination with 0.5 μM SAHA. ND = not determined * *P* ≤ 0.05, ** *P* ≤ 0.01, *** *P* ≤ 0.001. Statistical significance of BDZs alone is compared to DMSO (blue asterisks). When combined with SAHA, statistical significance is compared to SAHA (orange asterisks).

**Figure 5 viruses-12-00191-f005:**
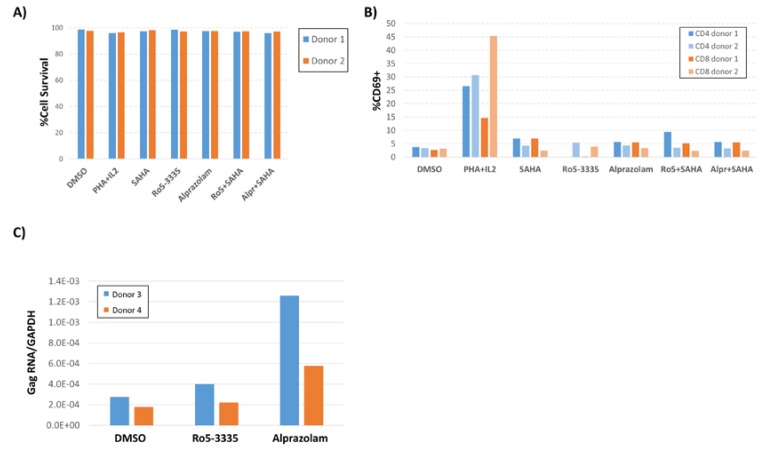
Effect of Ro5-3335 or Alprazolam on viability and HIV-1 transcription in PBMCs. Peripheral blood monocytes (PBMC) from two healthy donors were treated with 10 μM of Alprazolam or 50 μM of Ro5-3335 alone or in combination with 0.5 μM SAHA. 24 h post treatment, cells were stained with Zombie Yellow Live/Dead staining and then stained for CD3, CD4, CD8, and CD69. Then, cells were fixed for flow cytometry. (**A**) Percentage of live PBMCs after each treatment. (**B**) T-cell activation was graphed as the percentage of CD3+/CD4+ or CD3+/CD8+ cells that were CD69^+^. (**C**) PBMC from two HIV-1 positive subjects that were suppressed on antiretroviral therapy (ART) for more than 6 months were treated with Alprazolam or Ro5-3335 for 48 h. RNA was extracted using Trizol and cDNA was synthesized by RT reaction. qPCR was done to determine the relative increase in HIV-1 Gag mRNA as normalized to GAPDH. Panels A, B and C show the individual data from two human subjects.

**Figure 6 viruses-12-00191-f006:**
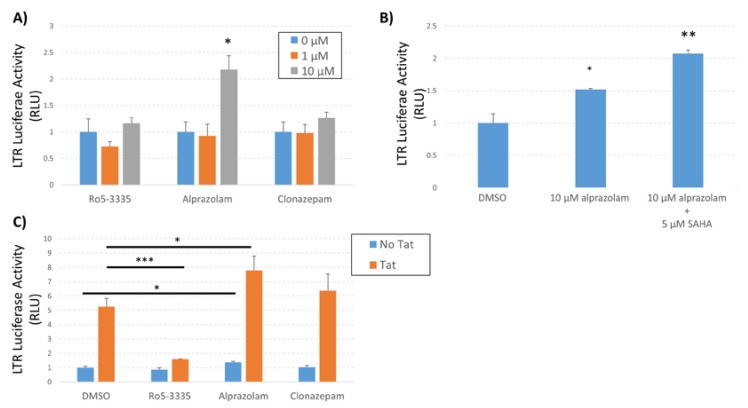
Effect of Alprazolam on HIV-1 transcription in TZMbl. TZMbl cells were seeded in 96-well plates and treated with (**A**) different concentrations of BDZs or (**B**) a combination of Alprazolam and SAHA. (**C**) TZMbl cells were transfected with pCMV-Tat or pUC19 control plasmid and treated with 10 µM of the indicated BDZ. At 48 h after treatment, the cells were lysed and equal mass of protein lysates were used to determine the production of luciferase from the integrated LTR. Results are shown as the relative luciferase activity compared to control. Each data point is the average of three replicates. * *P* ≤ 0.05, ** *P* ≤ 0.04, *** *P* ≤ 0.0001.

**Figure 7 viruses-12-00191-f007:**
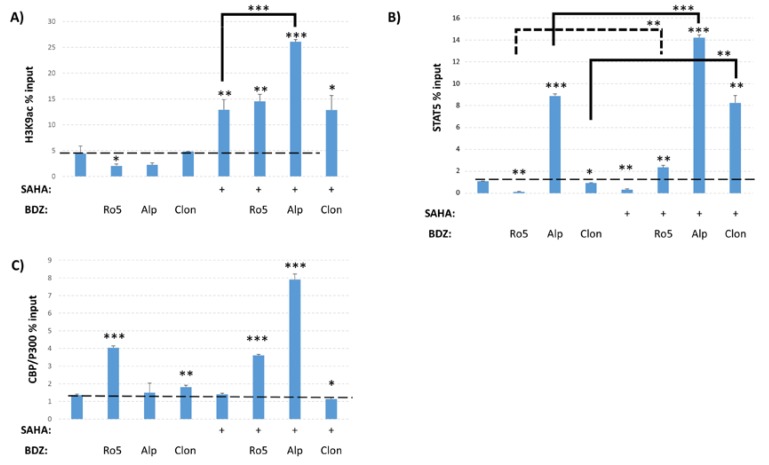
Epigenetic changes driven by Ro5-3335, Alprazolam, and Clonazepam. ChIP was performed to test the effect of Alprazolam and Clonazepam on occupancy of the HIV-1 LTR. TZMbl cells were treated with 10 μM Ro5-3335, Alprazolam, or Clonazepam in the presence or absence of 5 μM SAHA. ChIP was performed using antibodies against (**A**) H3K9ac, (**B**) STAT5, and (**C**) CBP/P300. Primers against HIV U3 LTR were used for qPCR. * *P* ≤ 0.05, ** *P* ≤ 0.01, *** *P* ≤ 0.001. Ro5: Ro5-3335, Alp: alprazolam, Clon: clonazepam.

**Figure 8 viruses-12-00191-f008:**
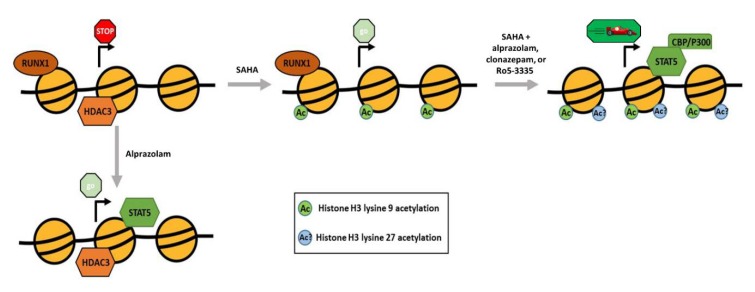
Model for the proposed effect of Alprazolam and SAHA on the HIV-1 LTR. A graphical representation of the nucleosomes (yellow circles) surrounding the transcription start site (arrow) in the integrated HIV-1 LTR. The association of proteins with the chromatin, acetylation state of the histones and relative amount of transcription are presented for each condition.

**Table 1 viruses-12-00191-t001:** List of tested clinically approved BDZs, a [42], b [43], c [44], d [45], e [46], f [47], g [48], h [49].

BDZ (Trade Name)	Common Clinical Uses	Peak Plasma Concentration (Dose Given)	Maximum Dose for Adults [39].
Alprazolam (XANAX)	Anxiolytic	0.0259 to 0.119 μM ^a^ (0.5 – 3.0 mg)	10 mg/day
Bromazepam (LECTOPAM)	Anxiolytic	0.44 μM ^b^ (12 mg)	---
Clobazam (ONFI)	Anxiolytic, anticonvulsant	0.73 to 2.3 μM ^c^ (20 mg)	40 mg/day
Clonazepam (KLONOPIN)	Anxiolytic, anticonvulsant	0.17 μM ^d^ (2 mg)	20 mg/day
Clorazepate (TRANEXENE)	Anxiolytic	1.3 μM ^e^ (20 mg)	90 mg/day
Diazepam (VALIUM)	Anxiolytic	1.16 μM ^f^ (10 mg)	40 mg/day
Estazolam (PROSOM)	Hypnotic	0.335 μM ^g^ (2 mg)	2 mg/day
Flunitrazepam (ROHYPNOL)	Hypnotic	0.047 μM ^h^ (2 mg)	---

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
