# Peer review of "Benzodiazepines Drive Alteration of Chromatin at the Integrated HIV-1 LTR"

_viruses, 2020, doi:10.3390/v12020191_

Round 1
Reviewer 1 Report
Following the demonstration that inhibition of RUNX1 transcription factor by Ro5-3335 benzodiazepine increased the ability of the histone deacetylase inhibitor SAHA to turn on HIV-1 LTR driven transcription, Elbezanti and coauthors further investigated the molecular implications of RUNX1 in HIV latency. Particularly, they reported that the inhibition of RUNX1 by Ro5-3335 accounted for alterations of integrated HIV-1 LTR chromatin state, significantly increasing the occupancy of STAT5 and CBP/P300 and the subsequent HIV transcription in the presence of SAHA. The confirmation of such interaction also with other benzodiazepines, some of which showed significant activation even in the absence of SAHA, suggests that these drugs might be repurposed for the activation of latent HIV-1 transcription through STAT5 recruitment. Overall, I think that the study is well designed, with robust experimental systems and sound results, and the manuscript is well written. Overall, I hence suggest to accept the manuscript after the following minor revisions.
Minor revisions
Lines 38-39: please add the corresponding references after the sentence “Recent studies propose a way to eradicate HIV-1 through activation of the silenced virus in resting 38 memory CD4+ cells through a “shock and kill” approach”
Line 53: please define the abbreviation CBF-β, as it is mentioned here for the first time
Line 91: please specify the meaning of PBMC at the first mention
Line 103: please specify the meaning of IL17 at the first mention
Lines 103, 115 and throughout the text: for sake of consistency, I would indicate all the hours in the same way, either as a number or in letters
Line 124: even if it is quite understandable, I would specify “HIV Gag”
Line 130: i think it should be “have been harvested”
Line 132: please use the abbreviation for FBS
Line 144: I suggest to rephrase as follows “All antibodies used were diluted 1:100, and included: Histone H3 (acetyl K9) antibody [AH3-120]-ChIP Grade (Abcam, 144 ab12179), Pierce p300/CBP (CREB-Binding Protein) antibody, etc
Lines 149-150: I suggest to report all primers pairs in the same format, either with or without F and R indications
Line 224: please define the abbreviation GFP, as it is mentioned here for the first time
Line 267: the closing bracket is missing
Lines 246, 284 and 320: in figure captions, please remove “The” before the word “effect”
Line 294: I would write “to induce transcription of integrated LTR promoters IN cells from HIV-1 infected individuals”
Line 384: please add a spacing between “complex” and “1”
Lines 386-387: I would suggest to write “mono-, di-, and tri-methylation”
Lines 408 and 411: should it be “Triazole”?
Line 414: please add a spacing after “transcription”
Figures 3 to 6 text: probably a slightly increase of the character size would allow the better interpretation of the data
Author Response
Thank you for your efforts in helping us to improve our manuscript. Please see the attachment for specific responses to your concerns.

Reviewer 2 Report
The manuscript by Elbezanti et al concerns a follow-up investigation on HIV-1 latency, namely a search or the molecular mechanism of RUNX inhibition of the integrated HIV-1 LTR and the antagonistic effect of the benzodiazepine Ro5-3335 on the process. In the paper, the authors have analysed the chromatin state of the integrated HIV-LTR using Ro5-3335, a compound known to inhibit RUNX and to stimulate transcription of the viral genome (a ‘latency reversing agent’), and other benzodiazepines, using ChIP analysis. The authors conclude that benzodiazepines act as RUNX1 inhibitors, resulting in the recruitment of STAT5 and other positive factors to the LTR with concomitant alteration of the chromatin structure to increase proviral transcription and end latency.
Most experiments appear quite straightforward, but I have some problems with the results shown in Fig. 5, which concern the effect of Alprazolam on HIV-1 transcription in TZMbl cells. The authors conclude from the experiment that ‘Alprazolam does not act as an inhibitor of Tat’, which R05-3335 does (lines 317-318). However, when adding proper controls to the panels in Fig. 5, as should have been done, so, comparing the –Tat results from panel B to the +Tat results from panel C, it can be calculated that without inhibitor Tat increases transcription 5×, and with inhibitor R05-3335 transcription increases only (approximately) 1.5 × , but in the presence of Alprazolam and Clonazepam transcription by Tat increases 4× and 3×, respectively, suggesting that these two compounds also inhibit Tat, albeit less efficiently. Could the authors comment on that? Also, in panel A, do the blue bars, labelled ‘0μM inhibitor’, contain DMSO, as in the negative controls in panels B & C? In lines 327-328, the authors state that ‘Alprazolam may have a different mechanism of action than Ro5-3335’, but is that completely true (see my comments above), also as the results from the J-Lat cells shown in Fig. 3 could be messed-up by a presumed expression of Tat in these cells as J-Lat cells contain a complete provirus, albeit with a frame-shift in env.
Other comments:
Line 39: I would change ‘virus’ into ‘provirus’ to make it clear what is analysed
Lines 74-75: is that completely true (see above)?
Lines 126-127: please give the expected size of the PCR fragment
Line 159: ‘in a patient samples’, please delete the a or the s
Line 160: ‘was doing’ could probably be rewritten in a more scientific way
Line 199: ‘cells were harvested…’
Line 208-210: something probably went wrong with the references here, as those cited have no connection with the research mentioned (e.g. ref. 27 is not by the Liu group, similarly refs 41 and 42 are not about the characterization of Ro5-3335 or a clinical trial, respectively). Please check throughout the manuscript.
Lines 235-236: probably due to the assay? Not due to a SAHA-independent mechanism for Alprazolam and Diazepam?
Fig. 3: why was Diazepam not tested here? Please explain
Fig. 7: what is the meaning of the sign apparently showing bananas in the third cartoon on the right?
Author Response

(The authors gave the same response as above.)

Reviewer 3 Report
The manuscript “RUNX1 inhibition drives alteration of chromatin at the integrated HIV-1 LTR” by Weam Elbezanti, Angel Lin, Alexis Schirling, Alexandria Jackson, et al., describes the molecular mechanisms involved in the induction of HIV-1 LTR driven transcription by RUNX1 inhibition using different benzodiazepine (BDZs) compounds. The results indicate Ro5-335, Alprazolam or Clonazepam treatment synergizes with SAHA to recruit STAT5 to the HIV LTR thereby activating HIV-1 transcription.
Here are my comments:
Majority of the data is derived from TZMbl cells that are derived from HeLa cells. Results obtained from monocytes and T cells (U1, ACH2, JLat or PBMCs from HIV-1 infected patients on ART treatment) would be more meaningful, especially for studies described in Fig 5 and Fig 6. Check and correct spelling and grammar errors.Author Response
Thank you for your efforts in helping us to improve our manuscript. Please see the attachment for specific responses to your concerns.

Reviewer 4 Report
The manuscript untitled “RUNX inhibition drives alteration of chromatin at the integrated HIV-1 LTR” is in perfect adequation with the field of Viruses journal. The authors have demonstrated that the use of different BDZ drugs, in addition to Ro5-3335, is able to activate transcription mediated by the HIV-1 LTR and have shown that proteins, involved in chromatin modifications, are recruited at the LTR region. As indicated by the author, the ability to reactivate dormant HIV-1 genome is a great challenge to eradicate the viral reservoir and to understand the mechanisms by which the silencing and reactivation are regulated. This study provides us interesting new clues. However, I think some experiments could be performed to enhance the quality of the work and the message.
Major comments
1/ The current title strongly suggests the reader that the whole manuscript deals with the inhibition of Runx1 and its consequences on chromatin which is correct for the results in figure 1 however the rest of the figures mainly focuses on the effect of other BDZs on the LTR interacting proteins and on the ability to activate transcription but the direct link with Runx1 is not established. This is confirmed at the end of the discussion by the authors (lines 415-419).
I strongly suggest the authors to study the effect of these different BDZs on the expression of reporter gene controlled by a mutant version of the HIV-1 LTR promoter (mutation of the binding site of Runx1 in the U3 region) or by promoters not regulated by Runx1 (see Fig. 3 from Klase et al., 2014 in PLOS Pathogens). Such demonstration would give an answer about this missing link with Runx1 and the study would be more consistent with the proposed title. The authors may also modify the title.
2/ It would be better to present the ChiP analysis of the Figure 1A like in figure 6 and especially with error bars. The results of the figure 1B support the data of the figure 1A as a functional assay. What a shame that the experimental conditions are not the same between both assays. Indeed, low doses of SAHA are not sufficient to stimulate the expression of luciferase, however some proteins are recruited on the LTR such as CBP-P300 and Runx/CBF without allowing histone H3 acetylation. What about the higher concentrations such as 10µM which promotes luciferase expression. Same question about the Ro5-3335 drug: used alone, it should not allow the transcription (Klase et al 2014) but should modify the binding of some proteins on the LTR.
So, I highly recommend that the authors harmonize these two experiments under the same conditions in order to be able to better compare them and understand what happens both in terms of protein binding to the LTR and their consequences on the expression of the reporter gene.
3/ Why is the recruitment of CBP/P300 on the LTR not favored with a 5µM of SAHA treatment in fig.6C whereas it is with 0.25µM and 1µM in fig.1A?
Minor comments
- Please, develop in the introduction the function and the targets of STAT5
- Please, indicate the concentrations of SAHA and Ro5-3335 used in Fig.1C.
- Lines 229-231: The authors indicate that only Alprazolam and Diazepam treatment result in significant activation of the LTR and the remainder… were unable to activate the LRE above the background levels. This is not true for flunitrazepam and clobazam. This should be corrected.
- The authors did not explain the choice of the 3 BDZ: Alpra, Clona and clora.
- line 332: please mention this information sooner in the manuscript “H3K9ac as a known marker for transcriptional activation” with references.
- Fig.1: µM instead of uM; define Ro5 abbreviation in the figure legend
- legend of table 1: Reference [34] is not in the good format
- results of fig.4A is the average of 3 independent assays but the error bars are missing.
- Fig.5: µM instead of uM, concentration of SAHA is missing
- Resolution of fig.7 can be improved.
- some digital object identifiers (DOI) are missing in the reference list
Author Response

(The authors gave the same response as above.)

Round 2
Reviewer 2 Report
In the revised version of the manuscript, the authors have responded adequately to all my suggestions, so that it can now be accepted for publication.
However, I still have my doubts about the bananas in Fig. 8, whose meaning can be missed by many non-native speakers. Perhaps it would be better to use a racing car to indicate that the transcription is going up?
Reviewer 4 Report
The authors have answered all the points requested. the manuscript is much improved and can be published.